# *Pneumocystis* Exacerbates Inflammation and Mucus Hypersecretion in a Murine, Elastase-Induced-COPD Model

**DOI:** 10.3390/jof9040452

**Published:** 2023-04-07

**Authors:** Diego A. Rojas, Carolina A. Ponce, Adriel Bustos, Vicente Cortés, Daniela Olivares, Sergio L. Vargas

**Affiliations:** 1Instituto de Ciencias Biomédicas (ICB), Facultad de Ciencias de la Salud, Universidad Autónoma de Chile, Santiago 8910132, Chile; 2Programa de Microbiología y Micología, ICBM, Facultad de Medicina, Universidad de Chile, Santiago 8380492, Chile

**Keywords:** *Pneumocystis*, mucus, COPD, mucins, Muc5ac, Muc5b, hypersecretion, inflammation

## Abstract

Inflammation and mucus hypersecretion are frequent pathology features of chronic respiratory diseases such as asthma and COPD. Selected bacteria, viruses and fungi may synergize as co-factors in aggravating disease by activating pathways that are able to induce airway pathology. *Pneumocystis* infection induces inflammation and mucus hypersecretion in immune competent and compromised humans and animals. This fungus is a frequent colonizer in patients with COPD. Therefore, it becomes essential to identify whether it has a role in aggravating COPD severity. This work used an elastase-induced COPD model to evaluate the role of *Pneumocystis* in the exacerbation of pathology, including COPD-like lung lesions, inflammation and mucus hypersecretion. Animals infected with *Pneumocystis* developed increased histology features of COPD, inflammatory cuffs around airways and lung vasculature plus mucus hypersecretion. *Pneumocystis* induced a synergic increment in levels of inflammation markers (Cxcl2, IL6, IL8 and IL10) and mucins (Muc5ac/Muc5b). Levels of STAT6-dependent transcription factors Gata3, FoxA3 and Spdef were also synergically increased in *Pneumocystis* infected animals and elastase-induced COPD, while the levels of the mucous cell-hyperplasia transcription factor FoxA2 were decreased compared to the other groups. Results document that *Pneumocystis* is a co-factor for disease severity in this elastase-induced-COPD model and highlight the relevance of STAT6 pathway in *Pneumocystis* pathogenesis.

## 1. Introduction

Chronic obstructive pulmonary disease (COPD) is a common condition in the adult population associated to cigarette smoking and ageing [1]. Chronic inflammation has a central role in the pathophysiology of the disease leading to airway epithelium changes such as the increase in the number of goblet cells (mucus secreting cells) and hyperplasia of mucus glands that cause mucus hypersecretion and airway collapse, pulmonary emphysema with impaired oxygen exchange and fibrosis [1,2]. The long-term exposure of airways to several types of pollutants such as cigarette smoke favors the release of inflammation mediators such as TNFα, IL1β, IL8 and GM-CSF [3,4,5,6]. Macrophages are also activated under COPD conditions, releasing chemotactic factors such as CCL2, CXCL8 and CXCL9 [6] or inflammation mediators and cytokines such as TNFα, IL6, IL8, MCP-1, LTB4, reactive oxygen species (ROS) and metalloproteases such as MMP-9 and MMP-12 [7,8,9,10]. These molecules attract neutrophils (CXCL8), monocytes (CCL2) and Th1 cells (CXCL9) to the airways and infiltrate the region around bronchioles. It has been documented that CD4+ and CD8+ lymphocytes infiltrate the airways of COPD patients correlating positively with high alveolar destruction (emphysema) and airway obstruction [9]. Mucus hypersecretion added to damaged hair cells prevent the normal elimination of secretions and leads to mucus accumulation in the airways [11]. Pulmonary lesions in COPD patients have been reproduced in at least two murine models: the cigarette smoke model (CS) [12,13,14,15] and the elastase instillation model (ELT) [16,17,18]. In the CS model the animals are exposed to smoke long-term using variable intervals from twice a week for 4 weeks to 5 times per week for 6 months [12,13]. This exposure increases the expression of inflammation mediators and fibrosis [12]. However, lesions observed in severe COPD patients such as emphysema are not generated [14]. In addition, in animals which were stopped from exposure to smoke the COPD lesions did not progress, even returning to the previous state before cigarette smoke exposure [15]. Moreover, the ELT model has several advantages compared with the CS model. First, it can induce emphysema and it can generate several other lung damages [16]. In rats, the use of elastase induces changes in the alveolar wall leading to alveolar space enlargement [17,18,19]. The lesions are progressive and could be observed even after 6 months from elastase application [20].

It is well-recognized that viral and bacterial infections worsen COPD symptoms [21,22]. However, the role of pathogenic fungus in these exacerbations has been poorly studied. *Pneumocystis* is an environmental fungus causative of severe pneumonia in immunosuppressed patients [23] that infects specifically mammals in a specie-specific manner [24]. *Pneumocystis jirovecii* infects humans and *Pneumocystis carinii* infects rats [24]. This fungus is highly prevalent in infants [25]. *Pneumocystis* increases inflammation and mucus hypersecretion in immunocompetent murine models inducing a strong Th2 [26,27,28,29] or Th17 immune responses [30] that show shared features of chronic respiratory diseases such as COPD. *Pneumocystis* is highly prevalent in COPD patients and in some studies it is correlated with the severity of the disease even when the fungal load is low [31,32,33]. Interestingly, *Pneumocystis* correlates with severity of COPD in HIV models [34] or in non-HIV patients [35] even when the burden of *Pneumocystis* is low [31,32]. In addition, animal models of *Pneumocystis* and COPD are few. The observation of smoke-induced COPD in a murine model where *Pneumocystis* synergized with smoke in inducing severe inflammation needs further study as is comparable with the type of inflammation response seen in COPD patients [36]. This evidence supports the hypothesis that mild infection by *Pneumocystis* potentiates inflammation in respiratory diseases such as COPD directly or as a co-factor [37]. However, the mechanisms underlying this role are still unknown [37]. Therefore, we aimed to determine the role of *Pneumocystis* in the potentiation of inflammation and mucus secretion in a COPD host. To this end, histological and molecular characterizations were performed, and inflammatory and mucus-related markers measured in an elastase-induced COPD animal model.

## 2. Materials and Methods

### 2.1. Ethics

The animal model was conducted in the Faculty of Medicine of the University of Chile under the protocol CBA#1110 approved by the Institutional Committee for Care and Use of Animals (CICUA) (certification No. 19336-MED-UCH. Studies were conducted according to the Animal Protection Law 20,380 of Chile and the Guide for the Care and Use of Laboratory Animals (8th Edition, National Academies Press, Washington, DC, USA).

### 2.2. Animal Model

The role of *Pneumocystis* infection in the severity of COPD, was evaluated in Sprague Dawley female rats of 300 g weight. All animals were derived of a single colony, and experiments were conducted at the Animal Research Facility of the University of Chile. Facility and housing conditions were the same as described previously [27]. Groups of 4 rats per cage were maintained in a HEPA-filtered air system (Lab Products, Aberdeen, MD, USA). Induction of COPD phenotype was generated by the instillation of porcine pancreatic elastase (SIGMA, St. Louis, MO, USA) as described elsewhere [16,17,18,19]. Two animal groups (*n* = 8) were instilled with 150 μL of elastase (1 U per animal) and other two control groups (*n* = 8) were instilled only with 150 μL of saline buffer. To prevent bacterial infections, animals were treated with tylosine (1 g/L) in the drinking water during all the experiment. Four weeks after instillation, the animals were co-housed with *Pneumocystis carinii*-infected animals (PcP rats) for 1 week. Co-housing was performed in a 1:4 co-habitation ratio, per cage. The PcP -seeding rats were prepared by administration of betamethasone (2 mg/L) in the drinking water and tylosin (1 g/L) for 8 weeks as previously described [27]. Half of the experimental groups were treated with trimethoprim-sulfamethoxazole (TMP-SMZ; 80 mg TMP and 400 mg SMZ per 5 mL) 15 mL/L in the drinking water to prevent *Pneumocystis* infection. In summary, 4 experimental animal groups were generated: saline instillation-only (SALINE), saline instillation with Pc-infected animals co-housing (Pc), elastase instillation-only (ELT) and elastase instillation + Pc-infected animals co-housing (ELT-Pc). Survival rate was high in all each animal group leading to a final group size of: SALINE = 7, ELT = 5, Pc = 7, ELT-Pc = 6. All animals were euthanized 14 weeks after instillation, using deep anesthesia with ketamine and xylazine and exsanguination according to the scheme shown in Figure 1A. Lungs were extracted from the thorax and washed with sterile PBS. Left lobule was separated and fixed in 3.7% formalin buffered with PBS (pH 7.4). Right lobules were quickly frozen at −80 °C. These tissues were used for histology examination and determination of protein levels and gene expression.

### 2.3. Histology

Paraffin-embedded 5 μm thick sections obtained from formalin-fixed lungs immersed in formalin for 48 h, deparaffinized and stained with hematoxylin and eosin (H&E) or Alcian blue—periodic acid Schiff (AB/PAS) using standard methods. To evaluate inflammation, the proportion of peribronchiolar and perivascular cuffs was determined using a semi-quantitative scoring system [38]. In this system, at least 5 bronchioles and 5 vessels with <300 μm diameter per animal were evaluated. A score of 0 was assigned to an image with no surrounding cuffs; 1 was assigned to an image where cuffs were present in <25% of bronchioles or vessels; 2 was assigned to an image where cuffs were present in >50% and <75% of bronchioles and vessels; 3 was assigned to an image where >75% of bronchioles or vessels are surrounded by cuffs. AB/PAS-stained sections were used to determine bronchiolar epithelium occupied by mucus. The measurements were made using the software ImageJ (Version 1.53, NCBI, Bethesda, MD, USA). The percentage of stained area was determined as the ratio of AB/PAS-stained area and total epithelium area defined as the area between luminal surface and basal membrane. A total of 14 bronchioles per animal group were analyzed. All images were obtained using an Olympus BX60 microscope and the software Image Pro Plus version 5.1.0 (Media Cybernetics Inc., Rockville, MD, USA).

### 2.4. Detection of Pneumocystis

*Pneumocystis carinii* was detected by conventional PCR in total genomic DNA isolated from 20 mg of fresh-frozen lung tissue using the Genomic DNA Isolation Kit (Norgen Biotek, Thorold, ON, Canada). 200 ng of genomic DNA was used as template to amplify a portion of the *P. carinii* mitochondrial large subunit rRNA gene (mtLSU) following standard PCR conditions (initial denaturation step of 10 min at 95 °C and 38 cycles of 30 s at 95 °C, 30 s at 55 °C and 30 s at 72 °C). The sequences of the primers used in this detection are presented in Table 1 and were designed according to the specific *P. carinii* oligonucleotides pAZ102E and pAZ102H as previously described [39]. Actin was used as internal control. *Pneumocystis carinii* burden was determined using qPCR approach. To this end, *dhfr* gene primers were used to amplify a 223 bp PCR product that was compared with the amplification of the same product inserted in a pGEM-T Easy plasmid (Promega, Madison, WI, USA). qPCR experiments were performed using the specifications of the kit HOT FIREPol EvaGreen qPCR Mix Plus (Solis Biodyne, Tartu, Estonia) and the AriaMx qPCR instrument (Agilent, Santa Clara, CA, USA).

### 2.5. Gene Expression Determinations

To evaluate changes in mRNA levels of inflammation and mucus gene activity markers, total RNA was extracted from fresh-frozen tissue lungs. Extractions were performed using 20 mg of lung tissue, which were homogenized in the presence of 500 μL of RNA solv reactive (Omega Biotek, Norcross, GA, USA). Then, to eliminate organic molecules, reactions were mixed by vortex in the presence of 100 μL of chloroform. Organic and aqueous phases were separated by cold centrifugation for 15 min at 14,000× *g*. Aqueous phase was transfer to a clean microcentrifuge tube and mixed with 250 μL of isopropyl alcohol. Reactions were incubated at −20 °C for one hour. Then, RNA was collected by cold centrifugation for 15 min at 14,000× *g*. RNA pellets were washed by the addition of 70% ethanol. RNAs were suspended in 50 μL of nuclease-free water, quantified and stored at −80 °C. 2 μg of each RNA was incubated with 1 unit of DNAse (Thermo Fisher Scientific, Waltham, MA, USA) and their specific buffer in 10 μL final volume during 30 min at 37 °C. Then, 2 μL of 25 mM EDTA was added to the mix to inactivate the enzyme at 65 °C for 10 min. After DNAse treatment, each RNA was incubated during 5 min at 70 °C with 0.5 μg of random hexamer primers (Promega, Madison, WI, USA) in 12.5 μL final volume. Then, the reactions were kept in ice during 5 min and nucleotides, buffer, RNAse inhibitor and reverse transcriptase were added according to M-MLV protocol (Promega, Madison, WI, USA) until complete a final volume of 25 μL. Briefly, mixes were incubated during 90 min at 37 °C and to inactive the enzyme, an additional step of 10 min at 70 °C was added to the protocol. The resulting one-stranded cDNAs were diluted 1:4 and 2 μL were used as template in qPCR experiments. Primers used in this work are listed in Table 1. To amplify each gene marker, all experiments were performed using HOT FIREPol EvaGreen qPCR Mix Plus (Solis Biodyne, Tartu, Estonia) and the AriaMx qPCR instrument (Agilent, Santa Clara, CA, USA). PCR reactions were performed according to the following conditions: 12 min at 95 °C (initial hot start) and 40 cycles of 20 s at 95 °C, 30 s at 58 °C and 20 s at 72 °C. Actin was used as internal control to all gene markers. Results are expressed as fold change normalized by actin and refers to control group following the 2^−ΔΔCt^ method [40].

### 2.6. Protein Extractions

To evaluate changes in protein levels, proteins were extracted from 40 mg of fresh-frozen lung tissues. These tissues were homogenized in 500 μL of modified RIPA buffer (50 mM Tris pH 7.4, 1% NP-40, 0.5% sodium deoxycholate, 150 mM NaCl, 1 mM EDTA) supplemented with total protease inhibitor cocktail (Roche, Mannheim, Germany). Then, additional 300 μL of RIPA buffer and sodium dodecylsulfate at 0.01% final concentration were added to the extractions. Mixes were kept in ice for one hour. To disrupt long genomic DNA molecules, extractions were passed 3 times by a 21-gauge syringe. To collect proteins (supernatants), extractions were cold-centrifuged two times at 14,000× *g* for 15 min. Protein quantification was performed using Bradford Protein Kit (BioRad, Hercules, CA, USA). Samples of each lung tissue were prepared by the addition of laemmli buffer 1:1 ratio and stored at −80 °C.

### 2.7. Western Blotting Determinations

A total of 30 μg of each protein sample extract were analyzed to evaluate changes in the levels of specific mucin proteins. Proteins were separated by electrophoresis in a 0.5 cm thick 1.5% agarose gel prepared in the presence of TAE1X supplemented with 0.1% sodium dodecylsulfate. Samples were run until running front was situated at 4 cm from the wells. Then, proteins were transferred to a PVDF membrane (Amersham Biosciences, Amersham, UK) using a Mini Trans-Blot Electrophoretic Transfer Cell (BioRad, Hercules, CA, USA) during one hour at 350 mA. Then, the membrane was blocked with 5% non-fat milk in TBS during two hours at room temperature. Washed membranes were probed with anti-Muc5ac (sc-21701, Santa Cruz Biotechnology, Dallas, TX, USA), anti-Muc5B (sc-21768, Santa Cruz Biotechnology, Dallas, TX, USA) or anti-actin (sc-8432, Santa Cruz Biotechnology, Dallas, TX, USA) over night at 4 °C at 1:2000 dilutions in 1% non-fat milk prepared in TBS. Primary antibodies were detected using a secondary antibody conjugated with HRP enzyme activity (sc-516102, Santa Cruz Biotechnology, Dallas, TX, USA) and complexes were visualized chemiluminescence using the SuperSignal West PICO Plus kit (Thermo Fisher Scientific, Waltham, MA, USA). Protein bands were quantified using ImageJ software (Version 1.53, NCBI, Bethesda, MD, USA). Results were expressed according to the internal control actin protein.

### 2.8. Statistics

GraphPad Prism 9 (GraphPad Software Inc., San Diego, CA, USA) was used for all the analysis. Results were expressed as the mean ± standard deviation (SD). The final number of animals per group was 5 or 6. Distribution of data was evaluated by Shapiro–Wilk test and differences between experimental animal groups were evaluated using one-way ANOVA and Tukey’s multiple comparison test. A *p*-value < 0.05 was considered statistically significant.

## 3. Results

### 3.1. Pneumocystis Increase COPD-Features in an Elastase-Induced COPD Rat Model

Animals instilled with elastase (ELT) show a clear increment in the alveolar space compared with animals instilled only with saline (SALINE) (Figure 1A,B) and histological images show that the spaces observed in the animal group instilled with elastase and co-housed with PcP rats (ELT-Pc) were higher and more abundant than in the ELT group (Figure 1B) suggesting a synergic response between elastase and *Pneumocystis* infection. Perfusion fixation or inflation were not done precluding intergroup comparison using Mean Linear Intercepts. The infection status of the animals was determined by conventional PCR (Figure 1C). *Pneumocystis carinii* burden was determined using qPCR against the number of *dhfr* gene copies. No significant differences in the number of copies were detected in the infected groups with and without elastase instillation (Appendix A).

### 3.2. Pneumocystis and Elastase Increase the Number of Inflammatory Cuffs, without a Noticeable Additive Effect in this Elastase-Induced COPD Rat Model

Chronic inflammation is one of the hallmarks in COPD and occurs in conducting airways and vessels. To evaluate inflammation, H/E-stained lung sections were analyzed by microscopy under a 10× objective magnification and a semi-quantitative score approach described in materials and methods. The number of inflammatory cuffs observed around bronchioles and vessels increased in the elastase-instilled (ELT and ELT-Pc) and in the infected (Pc) groups when compared with those observed in the control group (Figure 2). However, no differences were detected between the experimental groups ELT, Pc and ELT-Pc (Figure 2B,C). These results suggest maximal stimulation response in these groups or that the frequency of inflammation cuffs around bronchioles and vessels is not potentiated by *Pneumocystis* in COPD-animals; however, the composition of the infiltrates observed in Figure 2A, such as size and number of cells, was different between experimental groups.

### 3.3. Pneumocystis Increases Inflammatory Markers in an Elastase-Induced COPD Rat Model

To relate the histological findings with molecular mechanisms and to identify differences between animal groups, mRNA levels of Tnf, Cxcl2, Il6, Il8, Il10 and GM-CSF were evaluated by qPCR. Proinflammatory cytokines Tnf, Cxcl2, Il6 and Il8 showed higher levels in the infected animal group compared with control and ELT group (Figure 3A–D). However, the level of mRNA observed in the ELT-Pc group was much higher, specially with Cxcl2, Il6 and Il8. Interestingly, anti-inflammatory Il10 showed high levels of mRNA in the ELT-Pc animal group compared with the other animals (Figure 3E). Levels of the cytokine GM-CSF did not show differences between the experimental groups (Figure 3F).

### 3.4. Pneumocystis Increases Mucus Secretion in the Airway Epithelium of Elastase-Induced COPD Rats

Mucus hypersecretion has been described as part of the features of many respiratory diseases including COPD. *Pneumocystis* infected lungs also increases mucus in the airway epithelium. To evaluate mucus secretion in these animals, AB/PAS-stained sections were evaluated under a microscopy using a 10× objective measuring 14 bronchioles per animal. Animals instilled and infected (ELT and Pc) showed an increment of mucus secretion in the airway epithelium compared with control (Figure 4A). However, the presence of mucus was much higher in the ELT-Pc animal group compared with ELT and Pc groups (Figure 4A). These observations were evident when stained sections were quantified. A significant increment of the mucus-stained area in the airway epithelium was observed in the ELT and Pc groups compared with control animals, whereas the ELT-Pc group showed a two-fold increment compared with ELT and Pc groups (Figure 4B). These results indicate that *Pneumocystis* could potentiate the mucus secretion response of the airway epithelium in COPD animal.

### 3.5. Pneumocystis Increases Mucin Levels in an Elastase-Induced COPD Rat Model

To relate the histological findings to mucus secretion, levels of mucins Muc5ac and Muc5b were evaluated by qPCR and western blot. mRNA levels of Muc5ac were 5, 25 and 45-fold higher than control animals in the experimental groups ELT, Pc and ELT-Pc, respectively (Figure 5A). mRNA levels of Muc5b were 1.5, 6 and 10-fold higher than control animals in the groups ELT, Pc and ELT-Pc, respectively (Figure 5B). In both cases, the increment in the ELT-Pc group was synergic. Protein levels of Muc5ac and Muc5b showed a synergic increment in the animals group ELT-Pc compared with ELT and Pc groups, where protein increments were observed compared with control animals (Figure 5C–F). These results indicate that *Pneumocystis* may induce a synergic mucus secretion response in COPD animals and complement accurately the observations in histological images.

### 3.6. Pneumocystis Modulates Synergically STAT6-Dependent Transcription Factors

To explore transcriptional activation as a probable mechanism that explains the previous findings associated to mucus secretion, mRNA levels of some transcription factors will be evaluated. There is evidence that relates *Pneumocystis* infection and STAT6 pathway induction leading mucus increment via transcriptional repression/activation of mucin genes [29,41]. Therefore, mRNA levels of STAT6 downstream transcription factors Gata3, FoxA2, FoxA3 and Spdef were evaluated to check the status of STAT6 pathway. Levels of Gata3 showed and synergic 5-fold increment in the group ELT-Pc compared with control group (Figure 6A). FoxA2 levels were decreased 1.5–2-fold in the animal groups ELT and Pc compared with control group. However, the group ELT-Pc showed a decrease of 4-fold compared with control animals (Figure 6B). In addition, levels of FoxA3 showed an increment of 5-fold in ELT and Pc groups and 25-fold in the group ELT-Pc compared with SALINE (Figure 6C) indicating the synergic effect of *Pneumocystis*. Similar results were observed in the mRNA levels of Spdef, where the increment of 1.5-fold of the Pc group was raised to 2.5-fold in the ELT-Pc animal group compared with control animals (Figure 6D).

## 4. Discussion

This work shows a synergic effect of the “mild” *Pneumocystis* infection of the immunocompetent host and elastase in increasing emphysema lesions, expression of inflammation markers and mucus secretion in an elastase-induced animal model of COPD such as lung damage infected with *Pneumocystis*. Moreover, in several aspects, the presented data in this work showed that the presence of *Pneumocystis* in the airways worsens features of COPD. The increment of inflammation and emphysema lesions is similar with previous reports described in COPD patients, where *Pneumocystis* colonization increases the severity of COPD disease in HIV and non-HIV patients [34,42], supporting the hypothesis that *Pneumocystis* is a co-factor in chronic inflammatory lung diseases [37]. Interestingly, a recent work documented the role of *Pneumocystis* in the development of COPD disease in immunosuppressed animals [43], indicating a novel function of this fungus in the origin of the COPD disease. The inflammation is a hallmark in COPD that leads to mucus hypersecretion, fibrosis and airway collapse due pulmonary emphysema that is associated to the severity of the disease [2]. The finding of an increment in inflammation markers in COPD animals infected with *Pneumocystis* support the previously described relation between this fungal infection and severity of disease described in patients and in animal models of cigarette-smoke-induced COPD [34,35,36]. *Pneumocystis* induces several pathways such as Th1 and Th2 immune responses and the NFkB pathway [30,41,42] which are associated to inflammatory processes. These pathways are also altered in COPD and could explain the synergy observed in elastase-instilled infected animals in the presence of *Pneumocystis*. A synergic increment in mRNA levels of several cytokines such as TNFα, Cxcl2, Il6 and Il8, which are modulated by the activation of NFkB pathway, was documented. These cytokines are upregulated via NFkB in the presence of pollutants, such as cigarette smoke, and are the most frequent COPD inductor [44,45]. Inflammation observed in COPD could be due to the NFkB-dependent recruitment of macrophages leading to the release of pro-inflammatory cytokines such as TNF, Cxcl2 and IL8 [3,46]. The increment of inflammatory markers has been described in *Pneumocystis* primary infection rat model [27] and are associated with the burden of *Pneumocystis* and sub-clinical lesions that could be related with the induction of new pathways associated to the increase in the severity of chronic pulmonary diseases. For example, an increase in GM-CSF levels is well-documented in COPD, supporting a role for alveolar macrophage recruitment in lung inflammation [47,48,49]. However, in this work, GM-CSF mRNA levels were not different between experimental groups. This may be consistent with mouse models where if a low elastase dose was applied a low number of alveolar macrophages was detected [50] or was necessary a multiple instillation protocol [51] and may be a negative feature of elastase-induced COPD animal model. The elastase dose used in this work was selected based on the animal species used in this model. Despite the increase in inflammatory cuffs around the airways and vessels, it was not different between elastase-instilled and infected experimental groups, the size of the cuffs and the cellular composition of the inflammatory infiltrates had quantifiable differences. The infiltrated tissue around bronchioles that has been characterized in COPD patients showed the presence of CD4+ and CD8+ lymphocytes correlating with histologic lesions suggestive of emphysema, and clear signs of airway obstruction and cell destruction due to the release of perforins and TNFα [9,52]. Interestingly, these immune cells are present in inflammation cuffs around the airways of *Pneumocystis* infected animals [26,27]. This evidence suggests common pathways for COPD and infection leading to inflammation processes. Additionally, a strong eosinophilic response has been associated to *Pneumocystis* infection in murine models [26,27,53] and is associated to a Th2 response. Interestingly, Th2 immune response has been one of the pathways described as induced during *Pneumocystis* infections and related to mucus secretion [27,29,41]. The increment in mucus secretion observed in airways of COPD animals infected with *Pneumocystis* is relevant due to the distal airways’ obstruction associated to mucus hypersecretion identified in other chronic pulmonary diseases [54]. The induction of mucus secretion associated to mild infection by *Pneumocystis* has been described as part of a STAT6/FoxA2 pathway immune response [29], leading to the increment of the levels of the two most relevant mucins of the airways: Muc5ac and Muc5b. In this work, we documented that both mucins are induced by *Pneumocystis* in COPD animals in a synergic form and, accordingly, mucus in airway epithelium was notoriously elevated in distal airways and documented histologically. The pathological features documented in distal airways may increase the resistance airflow by mucus obstruction impairing ventilation and in the most severe cases lead to a complete closure of the small airways [54]. Recent studies have documented the importance of Muc5ac and Muc5b in the initiation and progression of COPD [55,56], emphasizing the important role of mucus hypersecretion in the pathogenesis of COPD. Despite various pathways that could explain the increment of mucus in our animal model, evaluation of STAT6-dependent transcription factors was chosen because of previous data showing STAT6 pathway activation in *Pneumocystis* infection rat models [27,29,41]. The transcriptional repressor FoxA2 modulates negatively the expression of mucins genes and is under regulation of the STAT6 pathway [57]. In this work, mRNA levels of FoxA2 were synergically decreased in COPD infected animals leading to an increment of mucins expression. However, animal models of *Pneumocystis* primary infection show that only FoxA2 nuclear levels decreased in infected animals [29], suggesting that age of the animals and eventually other factors influencing the acquisition of *Pneumocystis* may influence the host response. For example, in primary infection by *Pneumocystis,* mRNA levels of IL10 [27] are not different between control and infected groups—whereas in our work levels of IL10 increased in infected adult animals indicating different inflammation responses and probably different signaling pathways induced. The other two transcription factors evaluated were the activators FoxA3 and Spdef, which are related to Th2 immune response and STAT6 pathway have been associated to mucus hypersecretion in chronic pulmonary disease [58,59]. FoxA3 and Spdef are overexpressed in airways of COPD patients [58]; however, there is no evidence of association between both transcription factors and *Pneumocystis* infection. Interestingly, in this work both transcription factors showed increased mRNA levels in COPD infected animals indicating a putative induction of the pathways leading to overexpression of FoxA3 and Spdef and supporting the hypothesis that STAT6 are induced in this animal model. Recently, a smoke cigarette-COPD induced mice model lacking Muc5b gene documented the mucus secretion dependent on STAT6 and Spdef transcription factors [60]. COPD has a strong Th1 response component, whereas *Pneumocystis* infection can induce several immune responses such as Th1, Th2 and Th17 [30,41,42]. Therefore, a mixed response in COPD animals infected with *Pneumocystis* could explain the synergic inflammatory and mucus secretion response documented in this work. Further characterization of other COPD-related pathways, such as the IL1β pathway, must be performed in this animal model to elucidate the novel role of *Pneumocystis* as co-factor in chronic pulmonary diseases.

The reported evidence of high circulation of *Pneumocystis* in the community [61] in COPD patients [32,33], and the accumulative evidence contributed by this and other animal models, demonstrates that this fungus may favor respiratory impairment and synergize with other insults that induce COPD warrant to continue these studies characterizing the role of *Pneumocystis* in chronic lung disease.

## Figures and Tables

**Figure 1 jof-09-00452-f001:**
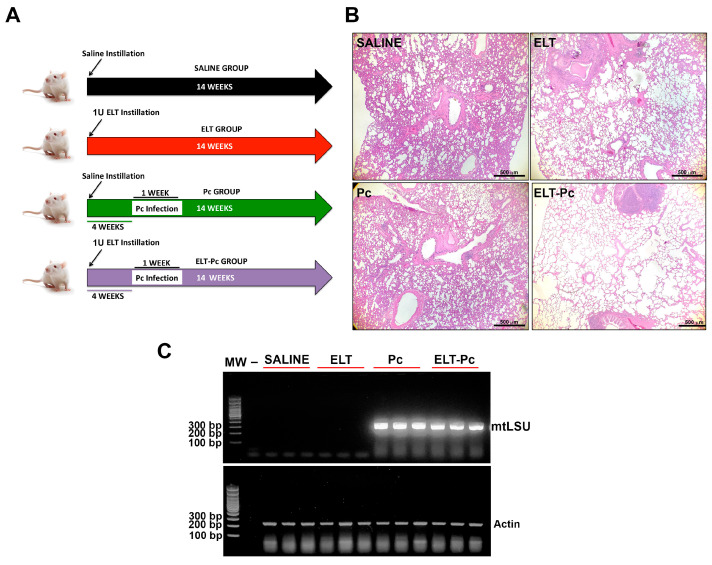
Evaluation of elastase-induced lung lesions. (**A**) Scheme of the animal model indicating the four experimental groups: Saline, Elastase (ELT), *Pneumocystis* infection (Pc), and Elastase-*Pneumocystis* infection (ELT-Pc). The experiment took 14 weeks total in all experimental groups. This time included the one-week of cohabitation with *Pneumocystis*-seeding rats starting 4 weeks after elastase instillation in the *Pneumocystis* groups. (**B**) Alveolar spaces were evaluated in lung sections from rats of the four groups using hematoxylin-eosin (H/E) staining. Microscopic analyses were performed using 40× magnification. (**C**) *Pneumocystis* infection was evaluated in the animals using PCR amplification of the mitochondrial large subunit (mtLSU) rRNA. Amplification of actin was used as internal control.

**Figure 2 jof-09-00452-f002:**
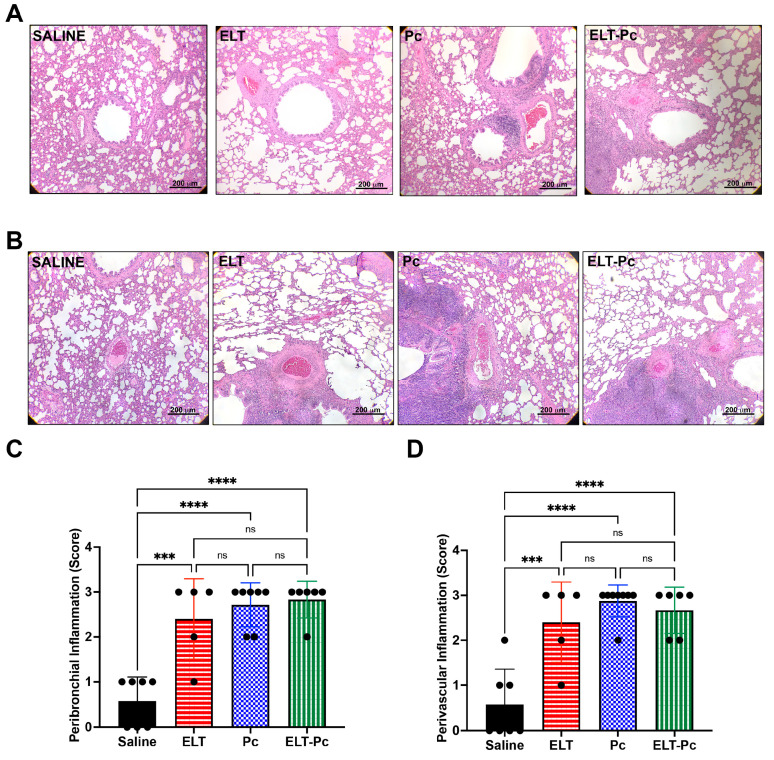
Evaluation of peribronchiolar and perivascular inflammation cuffs in the elastase-induced COPD model infected with *Pneumocystis*. Lung sections of rats from the four experimental groups were stained with hematoxylin-eosin and the presence of inflammation cuffs were identified by microscopic evaluation in images with 100× magnification. (**A**) Peribronchiolar and (**B**) perivascular inflammation cuffs were identified in the images. Quantification of the cuffs observed in the images was determined using a score described in the Materials and Methods section. Quantification of (**C**) peribronchiolar and (**D**) perivascular inflammatory cuffs is indicated. Data are expressed as mean ± SD and analyzed by ANOVA. ns = nonsignificant; *** = *p* < 0.001; **** = *p* < 0.0001.

**Figure 3 jof-09-00452-f003:**
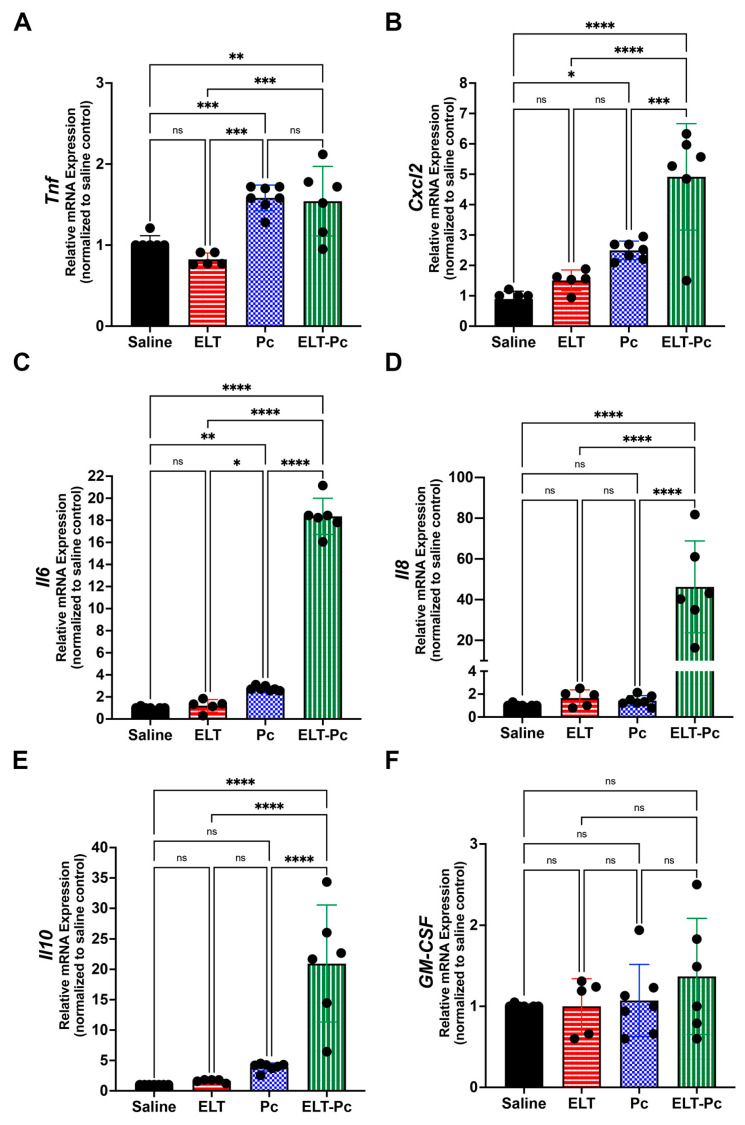
Characterization of cytokine profile in the elastase-induced COPD animal model infected with *Pneumocystis*. (**A**) *Tnf*, (**B**) *Cxcl2*, (**C**) *Il6*, (**D**) *Il8*, (**E**) *Il10* and (**F**) *GM-CSF* cytokines mRNA levels were evaluated by qPCR. Data were determined using the 2^−DDCt^, actin levels as internal control and normalized to saline control animal group. Data are expressed as mean ± SD and analyzed by ANOVA. ns = non-significant; * = *p* < 0.05; ** = *p* < 0.01; *** = *p* < 0.001; **** = *p* < 0.0001.

**Figure 4 jof-09-00452-f004:**
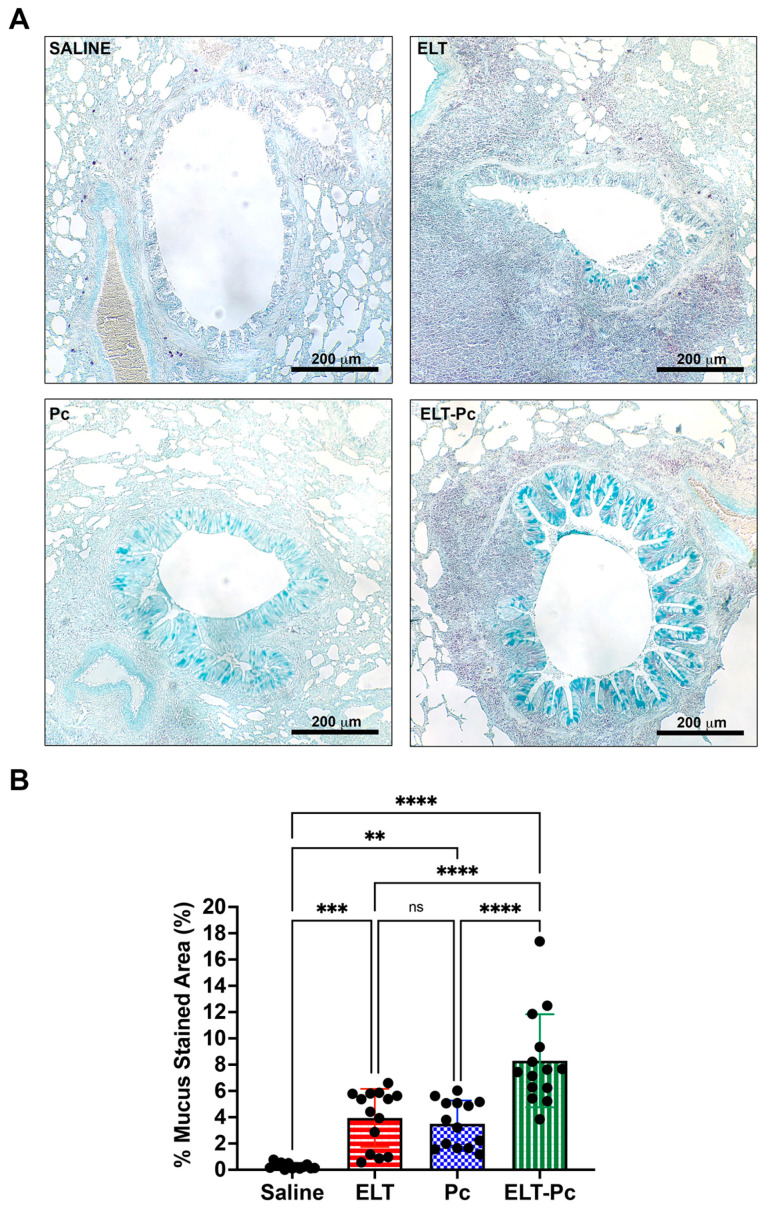
Evaluation of mucus secretion in the elastase-induced COPD animal model infected with *Pneumocystis*. (**A**) Representative images of lung sections from the four experimental animal groups stained with alcian-blue/periodic acid Schiff (AB/PAS). Microscopic magnification was 100×. (**B**) The stained area of the airway epithelium was quantified as described in the Materials and Methods section. Data are expressed as mean ± SD and analyzed by ANOVA. ns = non significant; ** = *p* < 0.01; *** = *p* < 0.001; **** = *p* < 0.0001.

**Figure 5 jof-09-00452-f005:**
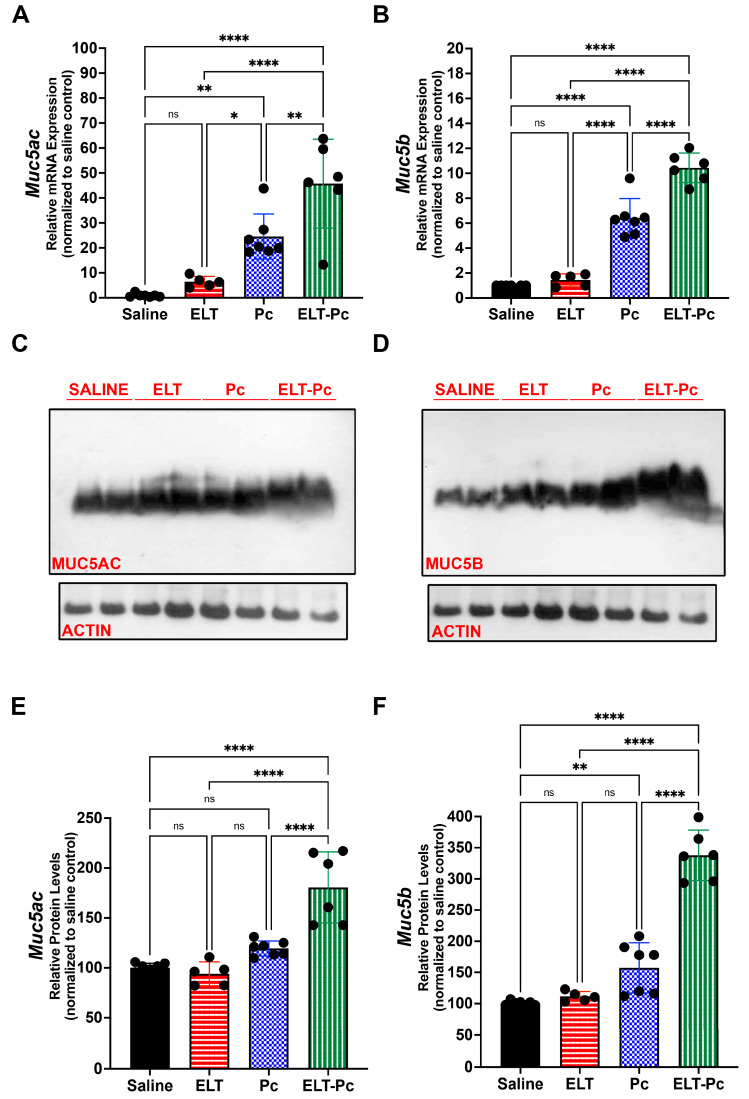
Evaluation of the mucin levels in the elastase-induced COPD animal model infected with *Pneumocystis*. mRNA levels of (**A**) *Muc5ac* and (**B**) *Muc5b* were evaluated by qPCR. Data were determined using the 2^−DDCt^, actin levels as internal control and normalized to saline control animal group. Data are expressed as mean ± SD and analyzed by ANOVA. ns = nonsignificant; * = *p* < 0.05; ** = *p* < 0.01; **** = *p* < 0.0001. Protein levels of the mucins were evaluated by western blot. Representative images of western blot to evaluate (**C**) Muc5ac and (**D**) Muc5b are indicated in the figure. Actin was used as internal control. Quantification of (**E**) Muc5ac and (**F**) Muc5b are indicated. Data are expressed as mean ± SD and analyzed by ANOVA. ns = nonsignificant; ** = *p* < 0.01; **** = *p* < 0.0001.

**Figure 6 jof-09-00452-f006:**
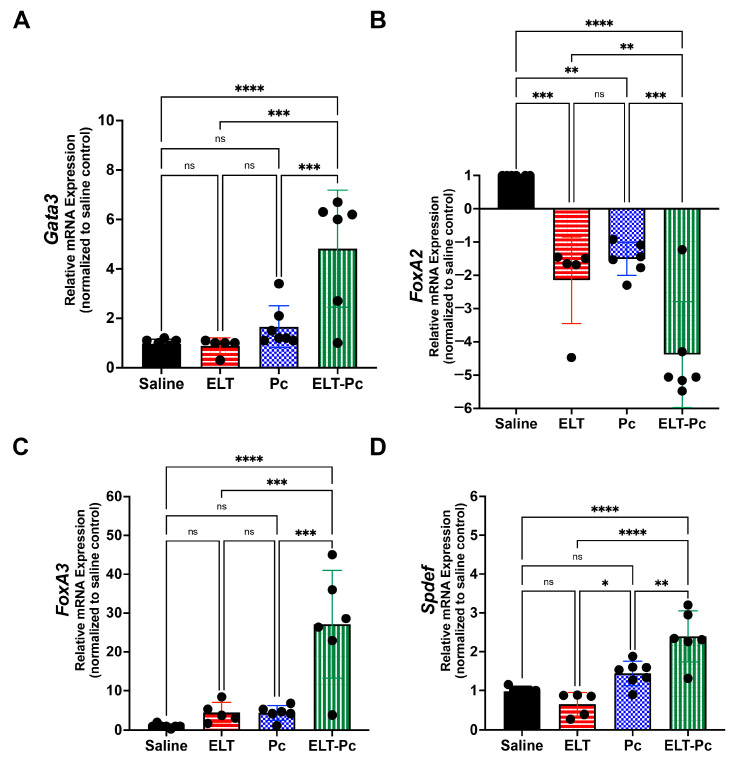
Evaluation of the expression level of STAT6-dependent transcription factor in the elastase-induced COPD animal model infected with *Pneumocystis*. mRNA levels of (**A**) *Gata3*, (**B**) *FoxA2*, (**C**) *FoxA3* and (**D**) *Spdef* were evaluated by qPCR. Data were determined using the 2^−DDCt^ method, actin levels as internal control and normalized to saline control animal group. Data are expressed as mean ± SD and analyzed by ANOVA. ns = nonsignificant; * = *p* < 0.05; ** = *p* < 0.01; *** = *p* < 0.001; **** = *p* < 0.0001.

**Table 1 jof-09-00452-t001:** Primers used in this work.

Gene		Sequence (5′–3′)	Size (bp)
pAZ102E (*mtSLU*)	Forward	GATGGCTGTTTCCAAGCCCA	227
pAZ102H (*mtSLU*)	Reverse	GTGTACGTTGCAAAGTACTC	
Dhfr11	Forward	GTTGCACTTACAACTTCTTATGG	223
Dhfr12	Reverse	TAGATCCAGAGATTCATTTCGAG	
*Actin*	Forward	CTTGCAGCTCCTCCGTCGCC	228
	Reverse	CTTGCTCTGGGCCTCGTCGC	
*Tnfa*	Forward	CAGCCGATTTGCCACTTCATA	71
	Reverse	TCCTTAGGGCAAGGGCTCTT	
*Cxcl2*	Forward	AACCATCAGGGTACAGGGGT	187
	Reverse	GGGCTTCAGGGTTGAGACAA	
*IL6*	Forward	CCCAACTTCCAATGCTCTCCTAATG	141
	Reverse	GCACACTAGGTTTGCCGAGTAGACC	
*IL8*	Forward	GAAGATAGATTGCACCGA	366
	Reverse	CATAGCCTCTCACACATTTC	
*IL10*	Forward	GGCTCAGCACTGCTATGTTGCC	116
	Reverse	AGCATGTGGGTCTGGCTGACTG	
*GM-CSF*	Forward	GCCATTGAGTTTGGTGAGGT	152
	Reverse	TCCTAAATGACATGCGTGCT	
*Muc5ac*	Forward	ACCACGGATATCAGAACCAGC	199
	Reverse	TGTCAAGCCACTTGGTCCAG	
*Muc5b*	Forward	CCTGAAGTCTTCCCCAGCAG	202
	Reverse	GCATAGAATTGGCAGCCAGC	
*Gata3*	Forward	CCATTACCACCTATCCGCCCTA	163
	Reverse	GTAGAGGTTGCCCCGCAGTT	
*FoxA2*	Forward	GAAGATGGAAGGGCACGAG	184
	Reverse	TGACATGTTCATGGAGCCTG	
*FoxA3*	Forward	AGTGGAGCTACTACCCGGAG	116
	Reverse	GGGAGGGTAGGGAGAGCTAA	
*Spdef*	Forward	CTTCACTACTGCGCCTCCAC	193
	Reverse	CCACCTGTGCGGAATCTTCA	

## Data Availability

Not applicable.

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
