# Peer review of "Pneumocystis Exacerbates Inflammation and Mucus Hypersecretion in a Murine, Elastase-Induced-COPD Model"

_jof, 2023, doi:10.3390/jof9040452_

Round 1

Reviewer 1 Report

The authors provide evidence that Pneumocystis infection exacerbates inflammation and mucus secretion in a rat model of elastase-induced COPD. The work is relevant in that P. jirovecii infection has been found to exacerbate the progression of COPD in human patients. The dual-hit model used for the study is appropriate and the conclusions are supported by the data. 

One aspect of the study that the authors need to address in the discussion is the timing of the experimental endpoints. As depicted in Figure 1, the uninfected ELT rats are assessed 14 weeks after ELT instillation. However, the ELT-Pc rats are assessed at 18 weeks after ELT instillation. Thus, it is possible that some of the differential effects in the ELT-Pc group could be related to the additional time post ELT. Do the authors have any data to suggest that the additional 4 weeks post ELT does not significantly affect the lung? The authors should at least acknowledge and discuss this possibility in the Discussion.

The authors present Pneumocystis infection data for the experimental rats that was obtained by conventional PCR of genomic DNA. This data gives a yes/no answer with regards to infection, but does not quantify the pathogen burden in the experimental animals. It is possible that ELT instillation alters either the growth rate of Pneumocystis or the ability of the immune system to control Pneumocystis growth. Either could result in a larger pathogen burden in the ELT-Pc rats, which could affect the observed outcomes with regards to mucus and cytokine production. Can the authors provide quantitative data? Again, this possibility should at least be acknowledged and discussed in the Discussion.

The authors use the general term Pneumocystis throughout the manuscript, but also mention P. carinii. For clarity, they should add a few sentences to explain that P. carinii infects rats, while P. jirovecii infects humans. 

Author Response

The authors provide evidence that Pneumocystis infection exacerbates inflammation and mucus secretion in a rat model of elastase-induced COPD. The work is relevant in that P. jiroveciiinfection has been found to exacerbate the progression of COPD in human patients. The dual-hit model used for the study is appropriate and the conclusions are supported by the data. underline

One aspect of the study that the authors need to address in the discussion is the timing of the experimental endpoints. As depicted in Figure 1, the uninfected ELT rats are assessed 14 weeks after ELT instillation. However, the ELT-Pc rats are assessed at 18 weeks after ELT instillation. Thus, it is possible that some of the differential effects in the ELT-Pc group could be related to the additional time post ELT. Do the authors have any data to suggest that the additional 4 weeks post ELT does not significantly affect the lung? The authors should at least acknowledge and discuss this possibility in the Discussion.

RESPONSE: Thank you for your comment. We apologize because the text was no clear. All experimental groups were euthanized the same day 14 weeks after instillation. The co-habitation time of the two experimental groups that were infected with Pneumocystis carinii is included in the 14 weeks of the experiment showed in the Figure 1A. We added this point to clarify the legend of Figure 1.

The authors present Pneumocystis infection data for the experimental rats that was obtained by conventional PCR of genomic DNA. This data gives a yes/no answer with regards to infection, but does not quantify the pathogen burden in the experimental animals. It is possible that ELT instillation alters either the growth rate of Pneumocystis or the ability of the immune system to control Pneumocystis growth. Either could result in a larger pathogen burden in the ELT-Pc rats, which could affect the observed outcomes with regards to mucus and cytokine production. Can the authors provide quantitative data? Again, this possibility should at least be acknowledged and discussed in the Discussion.

RESPONSE: Thank you for your suggestion. We have included a Supplementary Figure including the results of qPCR against Pneumocystis carinii DHFR gene indicating the burden of the fungus in the experimental animal groups. We did not find differences in the burden of the fungus between animals infected (with or without elastase instillation) so our results can not be explained using the fungus burden as a parameter.

The authors use the general term Pneumocystis throughout the manuscript, but also mention P. carinii. For clarity, they should add a few sentences to explain that P. carinii infects rats, while P. jirovecii infects humans.

RESPONSE: Thank you for the comment. We have added a sentence in the introduction section clarifying that Pneumocystis carinii is the fungus studied in this work can infects rats, whereas Pneumocystis jirovecii is the fungus that infects humans.

Reviewer 2 Report

This is an article that talks about the exacerbation of inflammation and mucus hypersecretion in a murine model of COPD.

The evidence presented is adequate for the conclusions presented in the article.

Quotations of some of the statements must be included in the document, for example in lines 30, 31, 48, 59.

Statement in line 65 says: “Pneumocystis is highly prevalent in COPD patients and correlates with the severity of the disease despite the burden of the fungus is low”, even when there is evidence of the aforementioned, the information in the literature in this regard is still not conclusive, so it should be said: "and in some studies, it is correlated with the severity of the disease even when the fungal load is low".

On lines 74 and 75, add the reference and that the clinical significance is unknown.

The statistical methodology and results are well presented and clearly described. Adequate to answer the research question and objectives.

In lines 338 to 342 they make an assertion based on two non-comparable pieces of evidence, since they compare their results with what is observed in clinical colonization processes, however, the experiment carried out is not with an animal model of colonization, rather it is a animal model of rats with COPD and infected with Pneumocystis, the article should adequately highlight the difference between colonization vs. infection, since several of the points mentioned in the discussion compare what is observed in animal models of infection with what is available as evidence regarding the role of Pneumocystis in colonization of immunocompetent patients with COPD.

The article presents important information on the COPD-Pnuemocystis relationship, which will serve as the basis for future research in animal models and clinical studies. It is recommended to adjust the mentioned points, especially regarding the wording of the discussion to ensure an adequate differentiation of the evidence presented in this model and what is known so far about the clinical effect of Pneumocystis (differences between colonization and infection).

Author Response

This is an article that talks about the exacerbation of inflammation and mucus hypersecretion in a murine model of COPD.

The evidence presented is adequate for the conclusions presented in the article.

Quotations of some of the statements must be included in the document, for example in lines 30, 31, 48, 59.

RESPONSE: Thank you for your comment. We have added references for each statement aforementioned.

Statement in line 65 says: “Pneumocystis is highly prevalent in COPD patients and correlates with the severity of the disease despite the burden of the fungus is low”, even when there is evidence of the aforementioned, the information in the literature in this regard is still not conclusive, so it should be said: "and in some studies, it is correlated with the severity of the disease even when the fungal load is low".

 RESPONSE: We changed the statement on line 65 according to your excellent suggestion.

On lines 74 and 75, add the reference and that the clinical significance is unknown.

 RESPONSE: Thank you for your comment. We have added the reference for the mentioned statement.

The statistical methodology and results are well presented and clearly described. Adequate to answer the research question and objectives.

 RESPONSE: Thank you very much for your comment.

In lines 338 to 342 they make an assertion based on two non-comparable pieces of evidence, since they compare their results with what is observed in clinical colonization processes, however, the experiment carried out is not with an animal model of colonization, rather it is a animal model of rats with COPD and infected with Pneumocystis, the article should adequately highlight the difference between colonization vs. infection, since several of the points mentioned in the discussion compare what is observed in animal models of infection with what is available as evidence regarding the role of Pneumocystis in colonization of immunocompetent patients with COPD.

 RESPONSE: We agree with Reviewer 2 in that colonization and infection (PcP) with Pneumocystis represents different degrees of severity that are highly dependent on the host response. Rats in this model were immunologically competent and their infection was the result of aerial contagion from cohabitation with immunocompromised animals with PcP. We compare our results with results obtained in animal models of colonization because they highlighted the similarity of responses. We added a few sentences in the discussion section to clarify this point indicating that the presence of Pneumocystis in the airways leads to worsen features of COPD. In addition we added a supplementary figure showing the quantification of the Pneumocystis dhfr gene copy number indicating that the burden is higher compared to other works (human colonization or rat primary infection) but lower compared with PcP data.

The article presents important information on the COPD-Pnuemocystis relationship, which will serve as the basis for future research in animal models and clinical studies. It is recommended to adjust the mentioned points, especially regarding the wording of the discussion to ensure an adequate differentiation of the evidence presented in this model and what is known so far about the clinical effect of Pneumocystis (differences between colonization and infection).

RESPONSE: We hope that the new version of the manuscript agrees with reviewer 2 suggestions